# HYPERCLICK: ADVANCING RELIABLE GUI GROUNDING VIA UNCERTAINTY CALIBRATION

## ABSTRACT

Autonomous Graphical User Interface (GUI) agents rely on accurate GUI grounding, which maps language instructions to on-screen coordinates, to execute user commands. However, current models, whether trained via supervised fine-tuning (SFT) or reinforcement fine-tuning (RFT), lack self-awareness of their capability boundaries, leading to overconfidence and unreliable predictions. We first systematically evaluate probabilistic and verbalized confidence in general and GUI-specific models, revealing a misalignment between confidence and actual accuracy, which is particularly critical in dynamic GUI automation tasks, where single errors can cause task failure. To address this, we propose HyperClick, a novel framework that enhances reliable GUI grounding through uncertainty calibration. HyperClick introduces a dual reward mechanism, combining a binary reward for correct actions with a truncated Gaussian–based spatial confidence modeling, calibrated using the Brier score. This approach jointly optimizes grounding accuracy and confidence reliability, fostering introspective self-criticism. Extensive experiments on seven challenge benchmarks show that HyperClick achieves state-of-the-art performance while providing well-calibrated confidence. By enabling explicit confidence calibration and introspective self-criticism, HyperClick reduces overconfidence and supports more reliable GUI automation.

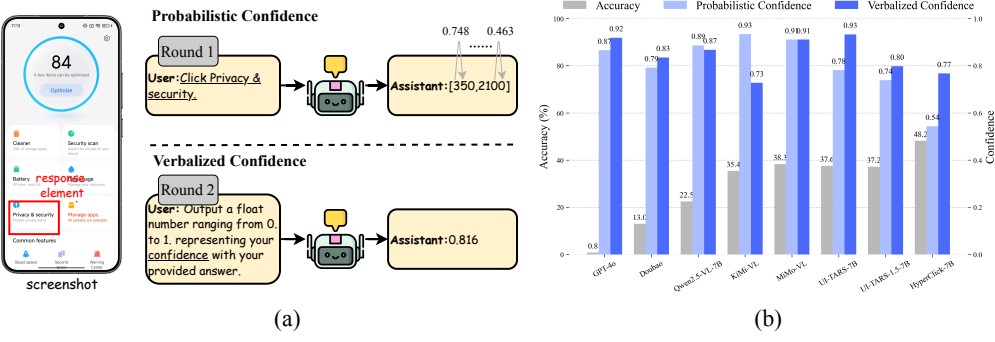

(a)                                   (b)

Figure 1: Overview of accuracy and confidence evaluation on ScreenSpot-Pro. (a): Illustration of probabilistic and verbalized confidence. Probabilistic confidence represents the probability of the model generating the next token corresponding to the target coordinates, while verbalized confidence indicates the model's self-reported certainty about its output in natural language. (b): Comparisons of accuracy, probabilistic confidence, and verbalized confidence for several general-purpose and GUI-specific models on the ScreenSpot-Pro benchmark. The models exhibit a higher confidence in their answers than in the accuracy that they actually achieve.

## 1 INTRODUCTION

The revolution of autonomous Graphical User Interface (GUI) agents is transforming human-computer interaction, enabling users to control mobile applications, web platforms, and complex desktop

software directly through natural language instructions (Wang et al., 2024b; Nguyen et al., 2024). At the heart of these agents lies the GUI grounding, the ability to accurately map textual commands to precise pixel coordinates on user interface elements (Cheng et al., 2024; Tang et al., 2025a). This fundamental task determines whether an agent can successfully execute user commands, making it the cornerstone of reliable GUI automation.

Recent progress in GUI grounding has been driven by supervised fine-tuning (SFT) with curated large-scale datasets (Wu et al., 2024; Gou et al., 2025; Xu et al., 2024) and reinforcement fine-tuning (RFT) with verifiable GUI-specific rewards (Lu et al., 2025; Luo et al., 2025; Liu et al., 2025b). Although these techniques yield strong performance, they share a critical weakness: the lack of self-awareness of their capability boundary, making it difficult to judge when predictions are reliable.

A reliable GUI agent should be aware of its limitations and accurately distinguish between what it can and cannot do (Ding et al., 2025). While this ability has been extensively studied in large language models (LLMs) (Xiong et al., 2023; Tian et al., 2023), it remains underexplored in GUI agents. The reliability level of an agent is assessed by the alignment between its confidence and actual performance (Ding et al., 2025). In this paper, we first evaluate probabilistic and verbalized confidence for several general models (OpenAI, 2024; Bai et al., 2025; Guo et al., 2025b; Team et al., 2025; Xiaomi, 2025) and GUI-specific models (Qin et al., 2025) on the ScreenSpot-Pro benchmark (Li et al., 2025), which emphasizes high-resolution displays, smaller target sizes, and complex environments. Specifically, probabilistic confidence reflects token-level likelihoods for predicted coordinates (Guo et al., 2017; Desai & Durrett, 2020), while verbalized confidence captures self-reported certainty in natural language (Lin et al., 2022; Yang et al., 2024b).

As shown in Figure 1, the models exhibit a higher confidence in their answers than in the accuracy that they actually achieve. In other words, even on challenging tasks, these agents remain overconfident in their predictions both probabilistically and from a self-assessed perspective. We argue that this is analogous to the hallucination problem commonly observed in LLMs and vision-language models (VLMs), where the model produces fluent, yet factually erroneous outputs while maintaining high confidence (Ji et al., 2023a;b; Kalai et al., 2025). This limitation is particularly critical in real-world GUI tasks, where their dynamic and continuous nature means that even a single error in an intermediate step can result in overall task failure.

To address this limitation, we propose HyperClick, a novel framework that enhances reliable GUI grounding through uncertainty calibration. Unlike prior approaches that treat grounding as a pure hit-or-miss classification problem, HyperClick explicitly integrates verbalized confidence estimation into the grounding process. Each prediction consists not only of a selected UI element, but also of a natural-language confidence statement, providing a self-assessment of reliability.

Specifically, we introduce two complementary rule-based reward mechanisms that optimize both action accuracy and uncertainty calibration. A binary reward enforces correct grounding actions, while a truncated Gaussian–based distribution models spatial confidence over the entire screenshot. The predicted confidence is then calibrated against this distribution using the Brier score (Glenn et al., 1950; Damani et al., 2025). This dual mechanism enables HyperClick to achieve two intertwined goals: accurate GUI grounding and well-calibrated confidence. More importantly, it fosters a form of introspectiveness, where the model not only acts but also critiques its own reliability. This self-criticism capacity reduces overconfidence, supports safer decision-making, and gradually expands the agent's boundaries of reliable operation.

Our contributions are summarized as follows:

- We systematically reveal that existing GUI grounding models are prone to overconfidence, analogous to hallucinations in LLMs and VLMs, and highlight their critical implications for reliable GUI automation.

- We propose HyperClick, the first GUI grounding framework that explicitly integrates uncertainty calibration, introducing a dual reward mechanism that jointly optimizes grounding accuracy and confidence reliability via binary correctness and truncated Gaussian–based confidence modeling.

- Through extensive evaluations on challenging GUI grounding benchmarks, HyperClick not only achieves state-of-the-art (SOTA) accuracy but also establishes well-calibrated confidence, enabling introspective self-criticism and more reliable GUI agents.

## 2 RELATED WORK

### 2.1 GUI AGENTS AND GROUNDING

GUI agents, as autonomous intelligent systems specialized in interacting with graphical user interfaces, have emerged as a key technology to automate complex desktop and mobile tasks (Wang et al., 2024b; Nguyen et al., 2024; Zhang et al., 2024). Recently, VLM-based GUI agents (Cheng et al., 2024; Wu et al., 2024; Qin et al., 2025) have demonstrated strong GUI comprehension by integrating visual perception with language reasoning, allowing them to handle diverse interface styles across applications. At the heart of VLM-based GUI agents lies the task of GUI grounding, which bridges natural language instructions with precise interface elements, thereby underpinning reliable GUI automation.

Early works (Cheng et al., 2024; Lin et al., 2025; Yang et al., 2024a) primarily focused on acquiring GUI-specific capabilities by collecting large-scale GUI corpora for SFT, thereby developing models customized for GUI tasks. SeeClick (Cheng et al., 2024) first introduced VLM to complete GUI tasks with only visual inputs. OS-Atlas (Wu et al., 2024), UGround (Gou et al., 2025), and Aguvis (Xu et al., 2024) aim to enhance perception by fine-tuning pre-trained models on a dataset constructed from diverse environments. UI-TARS (Qin et al., 2025) develops a native end-to-end GUI agent through large-scale GUI screenshots to enhance perception and reasoning for unified action modeling across platforms.

With the success of DeepSeek-R1-Zero (Guo et al., 2025a), RFT has drawn increased attention in the GUI-specific domain. UI-R1 (Lu et al., 2025), GUI-R1 (Luo et al., 2025), InfiGUI-R1 (Liu et al., 2025b), and BTL-UI (Zhang et al., 2025b) naively replicate techniques from DeepSeek-R1, prompting the model to think before generating an answer and optimizing the policy model with Verifiable GUI-specific reward functions. However, these native R1-based GUI agents overlook an important insight: Chain-of-Thought (CoT) reasoning degrades performance in GUI grounding, where precise spatial perception matters more than deep reasoning. Subsequently, GUI-G1 (Zhou et al., 2025) revisits the limitations of current R1-based GUI agents by introducing controllable box-size rewards for grounding tasks. SE-GUI (Yuan et al., 2025) proposes self-evolution approaches and continuous rewards to guide model learning. GUI-G$^2$ (Tang et al., 2025a) further introduced Gaussian reward modeling for GUI grounding. However, existing GUI grounding approaches primarily focus on improving grounding accuracy, while largely overlooking the importance of confidence calibration.

### 2.2 UNCERTAINTY CALIBRATION

The concept of uncertainty originates from the error analysis theory, where it quantifies the degree of confidence associated with a measurement (Oberkampf et al., 2002). This notion has been widely adopted in computer vision tasks such as object detection (Ren et al., 2015; Redmon et al., 2016) and semantic segmentation (Long et al., 2015; He et al., 2017), helping to assess the reliability of model predictions. With the rise of large language models (LLMs) and vision-language models (VLMs), several representative types of confidence signals have been proposed to capture the uncertainty of generated natural language: **(1) Probabilistic confidence** (Guo et al., 2017; Desai & Durrett, 2020), which uses token generation probabilities as a measure of uncertainty; **(2) Answer consistency confidence** (Zhang et al., 2023; Manakul et al., 2023; Fu et al., 2025), which quantifies uncertainty based on semantic consistency between multiple model outputs rather than token-level probabilities; and **(3) Verbalized confidence** (Lin et al., 2022; Yang et al., 2024b), where the model explicitly reports its confidence in natural language, providing an intuitive model-agnostic signal without requiring repeated sampling. Building on these advances, uncertainty estimation has been shown to improve the robustness and reliability of neural network systems by providing calibrated confidence for downstream decision-making.

## 3 METHOD

### 3.1 PROBLEM FORMULATION

GUI grounding can be formalized as the problem of mapping a natural language instruction to spatial coordinates corresponding to the target UI element on a given screenspot. From the perspective of

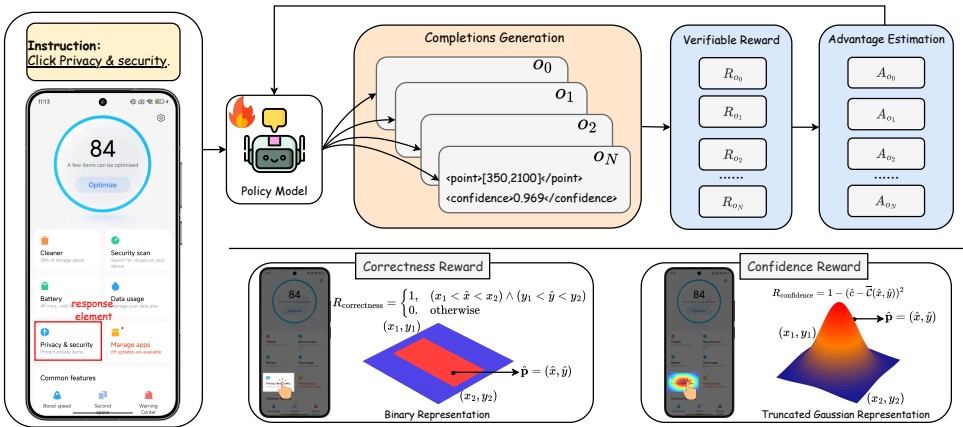

Figure 2: Framework of the proposed HyperClick, optimized with Group Relative Policy Optimization (GRPO). Given a screenshot and an instruction, the policy generates $N$ predictions, which are evaluated by a verifiable reward mechanism. The correctness reward measures grounding precision, while the calibration reward assesses uncertainty. For clarity, the reference model is omitted.

policy optimization, this task can be instantiated in two ways: location formulation (Wu et al., 2024; Tang et al., 2025a) and click formulation (Xu et al., 2024; Luo et al., 2025; Yuan et al., 2025).

- Location formulation: Given a screenshot $s$ and an instruction $q$, the policy model is optimized by predicting the bounding box $\hat{\mathbf{b}} = (\hat{x}_1, \hat{y}_1, \hat{x}_2, \hat{y}_2)$, where $(\hat{x}_1, \hat{y}_1)$ and $(\hat{x}_2, \hat{y}_2)$ denote the top-left and bottom-right corners of the UI element referred to by $q$.
- Click formulation: Alternatively, the policy model predicts a single point $\hat{\mathbf{p}} = (\hat{x}, \hat{y})$, corresponding to the center of the target element, which directly simulates a clicking action.

In this work, we adopt the click formulation as our primary paradigm, as it naturally aligns with executable actions in GUI interaction, simplifies the action space compared to bounding-box prediction, and provides a direct objective for reinforcement learning.

## 3.2 CONFIDENCE MODELING

Building on the introduction of the Gaussian distribution in error analysis theory (Gauss, 1809; 1877; MacKenzie, 1988) and recent advances in GUI-G$^2$ (Tang et al., 2025a), we model the confidence distribution in GUI grounding using a Gaussian formulation. Furthermore, as shown in Figure 2, since most UI element annotations are represented as bounding boxes, and to jointly account for correctness and confidence, we adopt a truncated Gaussian distribution (Galli et al., 1994) to model confidence.

**Truncated Gaussian Representation.** For each UI element with bounding box $\mathbf{b} = (x_1, y_1, x_2, y_2)$, the 2D Gaussian distribution on the screenshot interface can be denoted as:

$$\mathcal{N}(\mathbf{x}; \boldsymbol{\mu}, \Sigma) = \frac{1}{2\pi|\Sigma|^{\frac{1}{2}}}\exp(-\frac{1}{2}(\mathbf{x} - \boldsymbol{\mu})^{\mathrm{T}}\Sigma^{-1}(\mathbf{x} - \boldsymbol{\mu})), \tag{1}$$

where $\mathbf{x}$ means any point on the 2D interface, $\boldsymbol{\mu} = (\mu_x, \mu_y) = (\frac{x_1+x_2}{2}, \frac{y_1+y_2}{2})$ represents the center point of the UI element, and $\Sigma = \begin{pmatrix} \sigma_x^2 & 0 \\ 0 & \sigma_y^2 \end{pmatrix}$ is the covariance matrix. The diagonal structure assumes independence between the dimensions $x$ and $y$, simplifying the computation while maintaining expressiveness.

We preliminarily formulate the uncertainty distribution based on the constructed 2D Gaussian distribution on the interface. For each prediction point $\hat{\mathbf{p}}$ on the screenshot, the confidence value can

be computed:

$$\mathcal{C}(\hat{\mathbf{p}}) = 2\pi|\Sigma|^{\frac{1}{2}} \cdot \mathcal{N}(\hat{\mathbf{p}}; \boldsymbol{\mu}; \Sigma) = \exp(-\frac{1}{2}[\frac{(\hat{x} - \mu_x)^2}{\sigma_x^2} + \frac{(\hat{y} - \mu_y)^2}{\sigma_y^2}]). \tag{2}$$

For the center point $(\mu_x, \mu_y)$ of the grounding truth bounding box $\mathbf{b}$, the constructed value naturally reaches its maximum value of 1. This means that when the policy model predicts the point $(\mu_x, \mu_y)$, the model should have the highest confidence in its response.

Furthermore, we truncate the constructed confidence distribution by restricting it to the region defined by the bounding box $\mathbf{b}$, which aligns with the discriminative nature of the task. Specifically, confidence is assigned only when the predicted point $\hat{\mathbf{p}}$ is within $\mathbf{b}$; otherwise, the confidence is set to zero. In summary, the confidence distribution is modeled as a truncated Gaussian:

$$\overline{\mathcal{C}}(\hat{\mathbf{p}}) = \begin{cases} \mathcal{C}(\hat{\mathbf{p}}), & (x_1 < \hat{x} < x_2) \wedge (y_1 < \hat{y} < y_2), \\ 0, & \text{otherwise.} \end{cases} \tag{3}$$

**Adaptive Variance.** Previous approaches (Zhou et al., 2025; Tang et al., 2025a) have highlighted the difficulty bias in GUI grounding, where target elements with a smaller relative box size on the screenshot are more challenging. To handle UI elements with a wide range of sizes, we adopt the adaptive variance mechanism to control the confidence distribution on various platforms and screenshots:

$$\sigma_x = \alpha \cdot (x_2 - x_1), \quad \sigma_y = \alpha \cdot (y_2 - y_1), \tag{4}$$

which $\alpha$ is a scaling factor that controls the relative influence of the element size on the standard deviations.

### 3.3 TRAINING OBJECTIVE

**Correctness Reward.** As shown in Figure 2, we adopt the binary reward mechanism to guide the prediction point of the policy model $\hat{\mathbf{p}}$ within the bounding box $\mathbf{b}$. This discrete supervision directly aligns the policy objective with the success or failure of the grounding. Therefore, the correctness reward is expressed as follows:

$$R_{\text{correctness}} = \mathbb{1}_{\hat{\mathbf{p}} \in \mathbf{b}} = \begin{cases} 1, & (x_1 < \hat{x} < x_2) \wedge (y_1 < \hat{y} < y_2), \\ 0, & \text{otherwise.} \end{cases} \tag{5}$$

**Confidence Reward.** The purpose of the confidence reward is to encourage the policy model to evaluate and criticize the prediction generated $\hat{\mathbf{p}}$, making the confidence in the model output more precise. Thus, the confidence $\hat{c}$ of the model output should be aligned with the confidence distribution constructed in section 3.2. To achieve this, we introduce the Brier score (Glenn et al., 1950) to build the reward function, which can be thought of as a measure of the calibration of a set of probabilistic forecasts. The confidence reward can be formulated as follows.

$$R_{\text{confidence}} = 1 - (\hat{c} - \overline{\mathcal{C}}(\hat{x}, \hat{y}))^2. \tag{6}$$

This formulation provides several key properties. First, the closer the prediction confidence of the policy model $\hat{c}$ to the value corresponding to the constructed confidence distribution, the model will receive more reward. Second, when the model's prediction is incorrect and has a low confidence value for its generation, the policy model can still obtain a high confidence reward, which aligns with the model's motivation to self-criticize through confidence.

In summary, the final reward signal for the policy model combines a format reward $R_{\text{format}}$ with the correctness reward $R_{\text{correctness}}$ and the confidence reward $R_{\text{confidence}}$. The total reward is thus:

$$R = R_{\text{format}} + R_{\text{correctness}} + R_{\text{confidence}}. \tag{7}$$

We optimize HyperClick with Group Relative Policy Optimization (GRPO) (Shao et al., 2024), which extends the idea of relative advantage estimation to a group of predictions. Unlike standard policy gradient methods that rely on a single sampled return, GRPO leverages multiple candidate outputs to construct a relative reward signal, leading to more stable and informative optimization. Given $N$ generations $\{o_i\}_{i=1}^N$, each is evaluated by the reward function $R$. GRPO normalizes these rewards within the group to obtain relative advantages:

$$A_i = \frac{R(o_i) - \text{mean}(\{R(o_j)\}_{j=1}^N)}{\text{std}(\{R(o_j)\}_{j=1}^N)} \tag{8}$$

The training objective of GRPO is then defined as

$$\mathcal{J}(\theta) = \mathbb{E}_{\{o_i\}_{i=1}^N \sim \pi_{\theta_{\text{old}}}(\cdot|s,q)}$$

$$\frac{1}{N}\sum_{i=1}^{N}\left\{\min\left[\frac{\pi_\theta(o_i|s,q)}{\pi_{\theta_{\text{old}}}(o_i|s,q)}A_i, \text{clip}\left(\frac{\pi_\theta(o_i|s,q)}{\pi_{\theta_{\text{old}}}(o_i|s,q)}, 1-\epsilon, 1+\epsilon\right)A_i\right] - \beta \cdot \text{KL}(\pi_\theta||\pi_{\text{ref}})\right\}, \quad (9)$$

where $\pi_\theta$ denotes the policy model parameterized by $\theta$, $\epsilon$ is a hyperparameter that controls $\text{clip}(\cdot, 1 - \epsilon, 1 + \epsilon)$ and $\beta$ weights the KL regularization (Schulman et al., 2017; Shao et al., 2024) to stabilize training.

## 4 EXPERIMENTS

### 4.1 IMPLEMENTATION DETAILS

We implement HyperClick on top of Qwen2.5-VL-3B-Instruct and Qwen2.5-VL-7B-Instruct. Training data is sampled from multiple public GUI datasets, including OS-Atlas (Wu et al., 2024), Widget Caption (Li et al., 2020), UI-Refexp (Bai et al., 2021), and OmniAct (Kapoor et al., 2024), resulting in approximately 30K samples. Model training is conducted within the VLM-R1 (Shen et al., 2025) codebase. We train for one epoch on 16 NVIDIA H100 GPUs, using a learning rate linearly decayed from 1e-6 to 0 with a cosine scheduler, a global batch size of 16, 8 generations per instance, and a KL constraint coefficient of $\beta = 0.04$. To improve efficiency, we leverage FlashAttention-2 (Dao, 2023), adopt bfloat16 precision, and enable gradient checkpointing. During inference, the temperature is fixed to 0 to ensure reproducibility.

### 4.2 EVALUATION BENCHMARKS

We comprehensively evaluate the GUI grounding capability of HyperClick on ScreenSpot (Cheng et al., 2024) (SS), ScreenSpot-V2 (Wu et al., 2024) (SS2), ScreenSpot-Pro (Li et al., 2025) (SSP), MMBench-GUI (Wang et al., 2025) (MMG), UI-I2E-Bench (Liu et al., 2025a) (I2E), CAGUI (Zhang et al., 2025c) (CAG) and UI-Vision (Nayak et al., 2025) (UIV). More details about each evaluation benchmark are described in the Appendix.

### 4.3 MAIN RESULTS

**Comparisons with Baselines.** The main experimental results of HyperClick and comparisons with general models and GUI-specific models are shown in Table 4. HyperClick achieves consistently strong performance across all benchmarks. In particular, HyperClick-7B reaches new SOTA results in SS2 (93.7), SSP (48.2), MMG (79.6), I2E (76.5), CAG (82.9), and UIV (25.7), surpassing previous RFT-based approaches such as GUI-G$^2$ (Tang et al., 2025a) and SE-GUI (Yuan et al., 2025). In ScreenSpot (SS), HyperClick-7B obtains 91.5, which is highly competitive and comparable to the best results (92.0) of GUI-G$^2$. Moreover, HyperClick also demonstrates strong performance, outperforming much larger GUI-specific models, such as UI-TARS-72B (Wu et al., 2024) and Aguvis-72B (Xu et al., 2024).

A key source of HyperClick's improvement lies in the introduction of uncertainty calibration, which equips the model with a self-criticism mechanism. Unlike GUI grounding models that rely solely on sparse binary (Lu et al., 2025; Luo et al., 2025) or continuous (Yuan et al., 2025; Tang et al., 2025a) correctness rewards, HyperClick leverages a calibrated confidence distribution to explicitly distinguish between reliable and uncertain predictions. This enables the policy to penalize overconfident errors while reinforcing well-calibrated clicks. As shown in Table 4, such self-criticism translates into consistent gains across benchmarks, highlighting that calibrated confidence improves the model's ability to generalize across diverse UI environments. These results confirm that confidence-aware grounding not only enhances accuracy but also makes the model more robust to task difficulty and annotation variability.

**The confidence of HyperClick is reliable.** To evaluate whether HyperClick is truly reliable, we introduce the average precision (AP) of object detection (Lin et al., 2014), which adopts

Table 1: GUI grounding accuracy on seven benchmarks including ScreenSpot (Cheng et al., 2024) (SS), ScreenSpot-V2 (Wu et al., 2024) (SS2), ScreenSpot-Pro (Li et al., 2025) (SSP), MMBench-GUI (Wang et al., 2025) (MMG), UI-I2E-Bench (Liu et al., 2025a) (I2E), CAGUI (Zhang et al., 2025c) (CAG) and UI-Vision (Nayak et al., 2025) (UIV). **Bold** and underline indicate the best and second-best results. **The detailed experimental results on each benchmark are in the appendix**.

| Model | Size | SS | SS2 | SSP | MMG | I2E | CAG | UIV |
|---|---|---|---|---|---|---|---|---|
| *General Models* | | | | | | | | |
| GPT-4o (OpenAI, 2024) | - | 18.8 | 20.1 | 0.8 | 2.9 | - | 21.0 | 1.4 |
| Claude (Anthropic, 2024) | - | 83.0 | - | 17.1 | 4.7 | - | - | 8.3 |
| Qwen2-VL (Wang et al., 2024a) | 7B | 42.9 | - | - | - | 48.7 | - | 2.7 |
| Qwen2.5-VL (Bai et al., 2025) | 3B | 55.5 | 80.9 | 16.1 | - | 41.7 | - | - |
| | 7B | 84.7 | 88.8 | 26.8 | 33.9 | 53.8 | 59.6 | 0.9 |
| Intern3VL (Zhu et al., 2025) | 8B | 79.5 | 81.4 | - | - | - | - | - |
| | 38B | 85.6 | 88.3 | - | - | - | - | - |
| MiMo-VL (Xiaomi, 2025) | 7B | 87.2 | 90.5 | 41.9 | - | - | - | - |
| *GUI-specific Models (SFT)* | | | | | | | | |
| CogAgent (Hong et al., 2024) | 18B | 47.4 | - | 7.7 | - | - | - | 8.9 |
| SeeClick (Cheng et al., 2024) | 9.6B | 53.4 | 55.1 | 1.1 | - | 26.4 | - | 5.4 |
| Aria-UI (Yang et al., 2024a) | 25.3B | 82.4 | - | 11.3 | - | - | - | 10.1 |
| ShowUI (Lin et al., 2025) | 2B | 75.1 | 77.3 | 7.7 | 16.0 | 41.5 | - | 5.9 |
| UGround (Gou et al., 2025) | 7B | 73.3 | - | 16.5 | - | 16.5 | - | 8.8 |
| UGround-V1 (Gou et al., 2025) | 2B | 77.7 | - | - | - | 57.4 | - | 12.9 |
| | 7B | 86.3 | - | 31.1 | 65.7 | 70.3 | - | 23.2 |
| OS-Atlas (Wu et al., 2024) | 4B | 70.1 | 71.9 | 3.7 | - | 44.3 | - | - |
| | 7B | 82.5 | 84.1 | 18.9 | 41.4 | 58.6 | 57.2 | 9.0 |
| Aguvis (Xu et al., 2024) | 7B | 84.4 | - | - | 45.7 | 53.2 | 68.7 | 13.7 |
| | 72B | 89.2 | - | - | - | - | - | - |
| UI-TARS (Qin et al., 2025) | 2B | 82.3 | 84.7 | 27.7 | - | 27.7 | - | - |
| | 7B | 89.5 | 91.6 | 35.7 | - | 61.4 | 61.8 | 17.6 |
| | 72B | 88.4 | 90.3 | 38.1 | 74.3 | 73.7 | - | 25.5 |
| TongUI (Zhang et al., 2025a) | 3B | 83.6 | 85.5 | 18.0 | - | - | - | 15.4 |
| | 7B | 86.0 | 88.7 | 24.7 | - | - | - | 18.0 |
| GUI-Actor (Wu et al., 2025) | 2B | 86.5 | 88.6 | 42.2 | - | - | - | - |
| | 7B | 88.3 | 89.5 | 44.6 | - | - | - | - |
| JEDI (Xie et al., 2025) | 3B | - | 88.6 | 36.1 | - | - | - | 19.0 |
| | 7B | - | 91.7 | 39.5 | - | - | - | 25.2 |
| *GUI-specific Models (RFT)* | | | | | | | | |
| UI-R1 (Lu et al., 2025) | 3B | 83.3 | 85.4 | 17.8 | - | 58.5 | - | - |
| UI-R1-E (Lu et al., 2025) | 3B | 89.2 | 89.5 | 33.5 | - | - | - | - |
| GUI-R1 (Luo et al., 2025) | 3B | - | - | 28.6 | - | - | - | - |
| | 7B | - | - | 31.3 | - | - | - | - |
| InfiGUI-R1 (Liu et al., 2025b) | 3B | 87.5 | - | 35.7 | - | 69.7 | - | - |
| GUI-G1 (Zhou et al., 2025) | 3B | 90.3 | - | 37.1 | - | - | - | - |
| SE-GUI (Yuan et al., 2025) | 7B | 88.2 | 90.3 | 47.3 | - | - | - | - |
| LPO (Tang et al., 2025b) | 8B | - | 90.5 | - | - | - | - | - |
| GUI-G$^2$ (Tang et al., 2025a) | 7B | **92.0** | 93.3 | 47.5 | - | - | - | - |
| *Ours* | | | | | | | | |
| HyperClick | 3B | 88.5 | 90.6 | 41.3 | 71.4 | 71.8 | 81.0 | 19.6 |
| | 7B | 91.5 | **93.7** | **48.2** | **79.6** | **76.5** | **82.9** | **25.7** |

`confidence` $\in \{0.5, 0.75, 0.9, 0.95\}$ as the boundary positive for counting positive and negative samples. As shown in Table 2, HyperClick consistently maintains high AP across all thresholds, and as the confidence threshold increases, the AP also gradually increases, which indicates that the model not only makes accurate predictions but also assigns well-calibrated confidence scores, rather than overestimating or underestimating its certainty. Furthermore, compared to baseline models, HyperClick shows a clear margin of improvement, particularly in the high-confidence regime

Table 2: The evaluation of HyperClick on ScreenSpot-Pro is conducted under reliable and reproducible settings. The "Original" accuracy refers to the results reported in the corresponding papers or reproduced by subsequent studies, while the "Replicated" accuracy denotes our reproduction using the vllm-project (Kwon et al., 2023) with the official model weights. The observed performance gaps may stem from differences in prompt design or in whether unparsed outputs are included during evaluation.

| Model | Size | Accuracy | | $AP^{conf=50}$ | $AP^{conf=75}$ | $AP^{conf=90}$ | $AR^{conf=95}$ |
|---|---|---|---|---|---|---|---|
| | | Original | Replicated | | | | |
| GPT-4o (OpenAI, 2024) | - | 0.8 | 0.8 | 0.9 | 0.9 | 1.2 | 1.0 |
| Doubao (Guo et al., 2025b) | - | - | 13.0 | 13.6 | 15.8 | 21.2 | 21.5 |
| Qwen2.5-VL (Bai et al., 2025) | 7B | 26.8 | 22.5 | 24.9 | 24.9 | 24.8 | 24.7 |
| KiMi-VL (Team et al., 2025) | 16B | 34.5 | 35.4 | 34.8 | 34.8 | 25.8 | 40.6 |
| MiMo-VL (Xiaomi, 2025) | 7B | 39.9 | 38.3 | 29.5 | 28.9 | 28.8 | 30.0 |
| UI-TARS (Qin et al., 2025) | 7B | 35.7 | 37.6 | 37.5 | 37.5 | 37.4 | 39.3 |
| UI-TARS-1.5 (Qin et al., 2025) | 7B | - | 37.2 | 37.6 | 37.5 | 37.5 | 40.4 |
| HyperClick | 3B | - | 41.3 | **70.6** | **76.0** | **78.0** | 78.0 |
| | 7B | - | **48.2** | 61.3 | 64.6 | 71.2 | **78.7** |

Table 3: Ablation study of reward configurations.

| $R_{format}$ | $R_{corectness}$ | $R_{confidence}$ | **Acc(%)** |
|---|---|---|---|
| | ✓ | | 47.5 |
| ✓ | ✓ | | 47.7 |
| | ✓ | ✓ | 48.0 |
| ✓ | ✓ | ✓ | 48.2 |

Table 4: Ablation study of confidence.

| $\alpha$ | **Acc(%)** |
|---|---|
| 0 | 47.7 |
| 1/2 | 48.0 |
| 1/4 | 48.2 |
| 1/6 | 45.7 |

Table 5: Ablation of baseline.

| **Model** | **Acc(%)** |
|---|---|
| Qwen2.5-VL | 26.8 |
| HyperClick | 48.2 |
| MiMo-VL | 39.9 |
| HyperClick | 49.5 |

($AP^{conf=90}$ and $AP^{conf=95}$). This suggests that HyperClick is capable of self-criticizing its predictions: when the model outputs a high confidence score, the prediction is highly reliable; when the score is low, it effectively signals uncertainty. Such behavior is crucial for practical deployment in GUI automation, where wrong but overconfident predictions may lead to catastrophic task failures.

## 4.4 ABLATION STUDY

We conducted an ablation study on ScreenSpot-Pro to verify the effectiveness of key components of HyperClick.

**Reward Mechanism.** The results in Table 3 demonstrate the importance of combining correctness and confidence rewards. Using only the format or correctness reward yields relatively limited improvements (47.5% and 47.7%, respectively). Introducing the confidence reward alone already achieves stronger performance (48.0%), while the combination of correctness and confidence rewards further increases the precision to 48.2%. This validates our motivation that confidence calibration acts as a self-critical signal, discouraging overconfident errors and reinforcing reliable predictions.

**Confidence Modeling.** Table 4, investigates the effect of the adaptive variance factor $\alpha$, Without confidence modeling based on the truncated Gaussian distribution ($\alpha$=0), which means only binary confidence is used for uncertainty calibration. Therefore, when $\alpha$=0, the confidence reward is represented as:

$$R_{confidence} = 1 - (\hat{c} - \mathbb{1}_{\mathbf{p} \hat{\in} \mathbf{b}})^2, \tag{10}$$

the policy model reaches 47.7%, which is weaker than the truncated Gaussian variants. Moreover, we set $\alpha$ according to the principle $3\sigma$ of the Gaussian distribution. Take the x direction as an example, $k \cdot \sigma_x = \frac{1}{2}(x_2 - x_1)$, where $k \in \{1, 2, 3\}$ and subtract the scaling factor $\alpha \in \{\frac{1}{2}, \frac{1}{4}, \frac{1}{6}\}$. As shown in the results, while too large ($\alpha = \frac{1}{2}$) or too small ($\alpha = \frac{1}{6}$) variances lead to suboptimal performance. Specifically, when $\alpha = \frac{1}{2}$, the variance is too large and the Gaussian distribution is excessively truncated within the bounding box. As a result, the confidence mass is overly concentrated near the center, which weakens the model's sensitivity to the boundary regions of the element. In contrast, when $\alpha = \frac{1}{6}$, the variance is too small, leading to a distribution that is too truncated. Consequently, the confidence at the edge of the bounding box is nearly zero, making the calibration too strict and reducing the tolerance to minor prediction deviations. For comparison, $\alpha = \frac{1}{4}$ provides a balanced

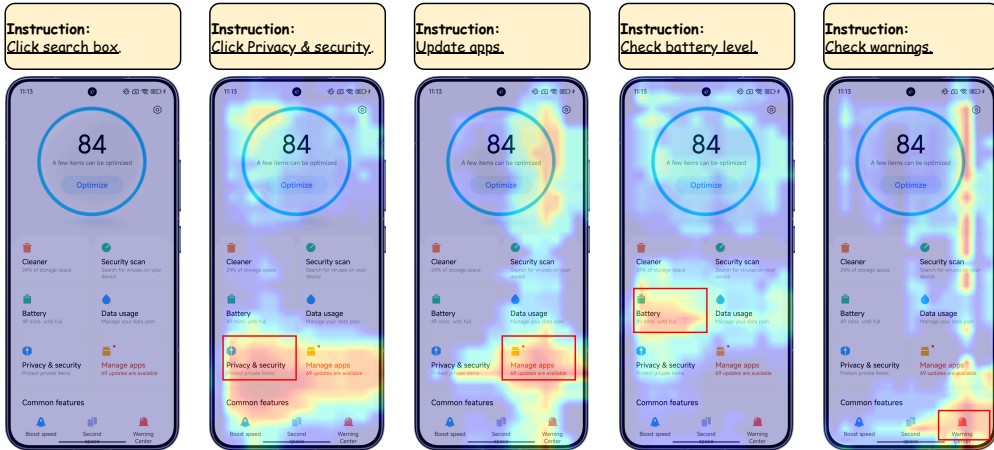

Figure 3: Visualization of the confidence distribution output by HyperClick. We inject the coordinates on the interface into the assistant's generation and enforce it to continue to output the confidence for the click position. The darker the color, the higher the confidence value.

trade-off between concentration and spread, providing the most effective uncertainty modeling and the highest precision (48.2%).

**Extension to other baselines.** As shown in Table 5, we further extend HyperClick to MiMo-VL (Xiaomi, 2025), a strong general-purpose VLM. With our training framework, MiMo-VL improves from 39.9% to 49.5%, demonstrating that HyperClick serves as a plug-and-play training paradigm for GUI grounding. Similarly, applying HyperClick to Qwen2.5-VL also brings substantial improvement (from 26.8% to 48.2%), confirming the generality and scalability of our approach across different foundation models.

## 4.5 VISUALIZATION

To better understand the effect of uncertainty calibration, we visualize the confidence distributions predicted by HyperClick in Figure 3. For each instruction, we inject the coordinates on the interface into the assistant's generation and enforce the policy model, continuing to output the click position. Thus, the heatmap represents its confidence in the possible click positions on the interface. We observe that the confidence is sharply concentrated around the ground-truth elements, while irrelevant regions exhibit low or near-zero confidence. This aligns with our design of truncated Gaussian modeling, where confidence only exists inside valid bounding boxes. Moreover, the adaptive variance mechanism adjusts the spread of the confidence distribution according to the element size: smaller UI elements yield tighter confidence peaks, whereas larger ones result in more diffuse heatmaps. These visualizations intuitively demonstrate how HyperClick achieves reliable and robust GUI grounding by avoiding overconfident but incorrect clicks.

## 5 CONCLUSION

In this work, we address the critical issue of overconfidence in GUI grounding models, which undermines the reliability of autonomous GUI agents. We introduce HyperClick, a novel framework that augments grounding with explicit uncertainty calibration. By combining binary correctness rewards with truncated Gaussian–based spatial confidence modeling, HyperClick enhances grounding accuracy while producing well-calibrated confidence estimates, enabling the agent to assess its own reliability introspectively. Extensive experiments on challenging benchmarks demonstrate that HyperClick achieves SOTA performance in both accuracy and calibration, substantially enhancing the reliability of GUI agents. Looking ahead, this framework can be extended to broader multimodal agentic settings, where reliable confidence estimation is essential for safe and reliable human-AI interaction.

ETHICS AND REPRODUCIBILITY STATEMENT

This research focuses on building a policy model for reliable GUI grounding. The data used are obtained by synthesizing or reprocessing previously released datasets, with all datasets or benchmarks properly cited. In this paper, there are no discrimination, bias, or fairness issues that need to be addressed. In addition, our models are not expected to generate potentially harmful content. To ensure reproducibility, we provide all experimental and data details in Section 4 and the corresponding appendices. We will release the source code and model checkpoints to support reproducibility.

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

# A  APPENDIX

## A.1  USE OF LLMS

In this paper, LLMs are employed solely as auxiliary tools for text refinement. Specifically, they are used to edit, polish and improve the clarity and readability of the manuscript, without contributing to the design of methods, the execution of experiments, or the analysis of results. All conceptual development, technical implementation, and empirical evaluation were independently conducted by the authors. The use of LLMs is therefore limited to linguistic enhancement, ensuring that the work's presentation is more precise and accessible to readers.

## A.2  LIMITATION

Although the effect of the uncertainty calibration mechanism proposed in this work has been verified, it has not been extended to GUI planning tasks. We believe that the reliability of planning is even more critical for the overall success of GUI automation, since inaccurate or overconfident planning decisions can propagate errors across multiple steps and ultimately lead to task failure. In future work, we plan to investigate how uncertainty calibration can be incorporated into planning modules, enabling agents to not only ground actions reliably but also make trustworthy high-level decisions throughout complex multi-step interactions.

## A.3  PROMPT

In this section, we detail the prompt for the replicated evaluation in ScreenSpot-Pro (Li et al., 2025). We follow the instructions they originally provided to reproduce and analyze the experimental results. The prompts are shown as follows:

> **GPT-4o's Prompt**
>
> Locate the UI element most related to the instruction {`problem`} on the screenshot. Output only a JSON in the format [{"point_2d": [...]}].

> **Doubao's Prompt**
>
> Locate the UI element most related to the instruction {`problem`} on the screenshot. Output only a JSON in the format [{"point_2d": [...]}].

> **Qwen2.5-VL's Prompt**
>
> Locate the UI element most related to the instruction {`problem`} on the screenshot. Output only a JSON in the format [{"point_2d": [...], "label": ... }].

> **KiMi-VL's Prompt**
>
> Point to the UI element most related to the instruction {`problem`} on the screenshot.

> **MiMo-VL's Prompt**
>
> Locate the UI element most related to the instruction {`problem`} on the screenshot. Output a JSON format [{"bbox_2d": [...], "label": ...}]./no_think

> **UI-TARS' and UI-TARS-1.5's Prompt**
>
> Point to the element related to the instruction {`problem`} on the screenshot.

Due to UI-TARS (Qin et al., 2025) and UI-TARS-1.5 (Qin et al., 2025) being trained with a large amount of GUI-specific data, the ability to follow instructions is relatively poor. To prompt such models to generate verbalized confidence in their predictions, we adopt a multi-round conversation to output confidence for their answer. Specifically, policy models use the above prompts for GUI grounding in the first round and in the second round, generate the verbalized confidence of the prediction according to the prompt below:

> **Confidence Prompt**
>
> Output only a float number ranging from 0 to 1, representing your confidence with your provided answer, without any format.

### A.4 STABILITY OF CONFIDENCE

To evaluate the reliability of HyperClick's confidence, we verified that the model's confidence in the same answer remains stable. We first let HyperClick predict the coordinates without doing a sample. Then, we inject the coordinates into the assistant's generation and instruct it to continue outputting confidence at a temperature of 1.0 for 8 times. As shown in Table 6, we report the mean variance for different sample sizes. The results indicate that both HyperClick-3B and HyperClick-7B maintain very low variance across different sampling scales, with the larger 7B model showing slightly more stable outputs. This suggests that the confidence estimation of HyperClick is well-calibrated, ensuring consistent reliability even under repeated sampling.

Table 6: Stability evaluation of the model for the same prediction.

| Model | Variance | | | | | |
|---|---|---|---|---|---|---|
| | 10 | 50 | 100 | 500 | 1000 | 1581 |
| HyperClick-3B | 0.020 | 0.028 | 0.023 | 0.020 | 0.020 | 0.020 |
| HyperClick-7B | 0.014 | 0.020 | 0.020 | 0.019 | 0.019 | 0.019 |

### A.5 DATA DETAILS

To provide a comprehensive grounding resource across diverse platforms, we construct a dataset containing 30K samples distributed across three representative domains: Mobile, Web, and Desktop. Each domain contains a balanced set of grounding instances that pair natural language commands with corresponding UI elements. The number of samples collected from each dataset is shown below.

Table 7: Statistics and sources of the grounding dataset adopted in HyperClick.

| Source | OmniAct | ShowUI-Web | UI-Refexp | Widgent-Caption | OS-Atlas | In-House |
|---|---|---|---|---|---|---|
| **Size** | 119 | 19172 | 280 | 3672 | 26114 | 1664 |

To construct high-quality samples for RFT, we first employ Qwen2.5-VL-7B (Bai et al., 2025) to generate raw data with the temperature set to 0, and identify cases where the model produces incorrect predictions. For each of these error cases, we then perform eight additional inferences with temperature 0.9 and extract the correctly predicted results as the final training data. In addition, prior to RFT, we incorporate an equal number of correctly predicted samples from Stage 1 to provide a cold start. This initialization not only stabilizes the training but also helps the model adhere to the target output format: `<point>[x,y]</point><confidence>conf</confidence>`.

### A.6 EVALUATION BENCHMARKS AND DETAILED EXPERIMENTAL RESULTS

In this section, we detail the benchmarks adopted in this work.

**ScreenSpot** evaluates GUI grounding across mobile, desktop, and web platforms. Provides a diverse set of interface types, enabling the comparison of model robustness across common user scenarios. Detailed experimental results and comparisons with baselines are shown in Table 8.

Table 8: GUI grounding accuracy on the ScreenSpot (Cheng et al., 2024) benchmarks over the Mobile, Desktop, and Web sub-tasks. **Bold** and underline indicate the best and second-best results.

| Model | Size | ScreenSpot v1 | | | | | | SSv1 Avg. |
| | | Mobile | | Desktop | | Web | | |
| | | Text (273) | Icon (229) | Text (194) | Icon (140) | Text (230) | Icon (206) | |
| --- | --- | --- | --- | --- | --- | --- | --- | --- |
| *General Models* | | | | | | | | |
| GPT-4o (OpenAI, 2024) | - | 30.5 | 23.2 | 20.6 | 19.4 | 11.1 | 7.8 | 18.8 |
| Claude (Anthropic, 2024) | - | - | - | - | - | - | - | 83.0 |
| Qwen2-VL (Wang et al., 2024a) | 7B | 61.3 | 39.3 | 52.0 | 45.0 | 33.0 | 21.8 | 42.9 |
| Qwen2.5-VL (Bai et al., 2025) | 3B | - | - | - | - | - | - | 55.5 |
| | 7B | - | - | - | - | - | - | 84.7 |
| InternVL3 (Zhu et al., 2025) | 8B | - | - | - | - | - | - | 79.5 |
| | 38B | - | - | - | - | - | - | 85.6 |
| *GUI-specific Models (SFT)* | | | | | | | | |
| CogAgent (Hong et al., 2024) | 18B | 67.0 | 24.0 | 74.2 | 20.0 | 70.4 | 28.6 | 47.4 |
| SeeClick (Cheng et al., 2024) | 9.6B | 78.0 | 52.0 | 72.2 | 30.0 | 55.7 | 32.5 | 53.4 |
| Aria-UI (Yang et al., 2024a) | 25.3B | 92.3 | 73.8 | 93.3 | 64.3 | 86.5 | 76.2 | 82.4 |
| ShowUI (Lin et al., 2025) | 2B | 92.3 | 75.5 | 76.3 | 61.1 | 81.7 | 63.6 | 75.1 |
| UGround (Gou et al., 2025) | 7B | 82.8 | 60.3 | 82.5 | 63.6 | 80.4 | 70.4 | 73.3 |
| UGround-V1 (Gou et al., 2025) | 2B | 89.4 | 72.0 | 88.7 | 65.7 | 81.3 | 68.9 | 77.7 |
| | 7B | 93.0 | 79.9 | 93.8 | 76.4 | 90.9 | 84.0 | 86.3 |
| OS-Atlas (Wu et al., 2024) | 4B | 85.7 | 58.5 | 72.2 | 45.7 | 82.6 | 63.1 | 70.1 |
| | 7B | 93.0 | 72.9 | 91.8 | 62.9 | 90.89 | 74.3 | 82.5 |
| Aguvis (Xu et al., 2024) | 7B | 95.6 | 77.7 | 93.8 | 67.1 | 88.3 | 75.2 | 84.4 |
| | 72B | 94.5 | 85.2 | 95.4 | 77.9 | 91.3 | 85.9 | 89.2 |
| UI-TARS (Qin et al., 2025) | 2B | 93.0 | 75.5 | 90.7 | 68.6 | 84.3 | 74.8 | 82.3 |
| | 7B | 94.5 | 85.2 | 95.9 | 85.7 | 90.0 | 83.5 | 89.5 |
| | 72B | 94.9 | 82.5 | 89.7 | 88.6 | 88.7 | 85.0 | 88.4 |
| TongUI (Zhang et al., 2025a) | 3B | 92.6 | 77.7 | 92.3 | 77.1 | 87.8 | 74.8 | 83.6 |
| | 7B | 91.9 | 79.5 | 93.8 | 80.0 | 89.1 | 81.6 | 86.0 |
| GUI-Actor (Wu et al., 2025) | 2B | 93.0 | 79.9 | 88.1 | 78.6 | 90.9 | 84.0 | 86.5 |
| | 7B | 94.9 | 82.1 | 91.8 | 80.0 | 91.3 | 85.4 | 88.3 |
| *GUI-specific Models (RFT)* | | | | | | | | |
| UI-R1 (Lu et al., 2025) | 3B | 95.6 | 84.7 | 90.2 | 59.3 | 85.2 | 73.3 | 83.3 |
| UI-R1-E (Lu et al., 2025) | 3B | 97.1 | 83.0 | 95.4 | 77.9 | 91.7 | 85.0 | 89.2 |
| GUI-R1 (Luo et al., 2025) | 3B | - | - | 93.8 | 64.8 | 89.6 | 72.1 | - |
| | 7B | - | - | 91.8 | 73.6 | 91.3 | 75.7 | - |
| InfiGUI-R1 (Liu et al., 2025b) | 3B | 97.1 | 81.2 | 94.3 | 77.1 | 91.7 | 77.6 | 87.5 |
| GUI-G1 (Zhou et al., 2025) | 3B | 98.6 | 85.8 | 96.4 | 80.7 | 91.4 | 82.3 | 90.3 |
| SE-GUI (Yuan et al., 2025) | 7B | - | - | - | - | - | - | 88.2 |
| GUI-G$^2$ (Tang et al., 2025a) | 7B | 96.7 | 90.8 | 95.9 | 88.6 | 90.9 | 86.9 | **92.0** |
| *Ours* | | | | | | | | |
| HyperClick | 3B | 96.7 | 83.9 | 92.8 | 80.7 | 88.7 | 83.5 | 88.5 |
| | 7B | 95.6 | 91.7 | 93.8 | 82.9 | 92.2 | 88.4 | 91.5 |

**ScreenSpot-V2** extends ScreenSpot with more challenging tasks and refined annotations. Additionally, it tests grounding accuracy in various real-world environments. Detailed experimental results and comparisons with baselines are shown in Table 9.

**ScreenSpot-Pro** focuses on high-resolution professional settings with expert-annotated tasks. Covers 23 applications, five industries, and three operating systems, making it one of the most comprehensive

Table 9: GUI grounding accuracy on the ScreenSpot (Cheng et al., 2024) and ScreenSpot-V2 benchmarks over the Mobile, Desktop, and Web sub-tasks. **Bold** and underline indicate the best and second-best results.

| Model | Size | ScreenSpot V2 | | | | | | SSv2 Avg. |
| | | Mobile | | Desktop | | Web | | |
| | | Text (290) | Icon (211) | Text (194) | Icon (140) | Text (234) | Icon (203) | |
| --- | --- | --- | --- | --- | --- | --- | --- | --- |
| *General Models* | | | | | | | | |
| GPT-4o (OpenAI, 2024) | - | 26.6 | 24.2 | 24.2 | 19.3 | 12.8 | 11.8 | 20.1 |
| Qwen2.5-VL (Bai et al., 2025) | 3B | 93.4 | 73.5 | 88.1 | 58.6 | 88.0 | 71.4 | 80.9 |
| | 7B | 97.6 | 87.2 | 90.2 | 74.2 | 93.2 | 81.3 | 88.8 |
| *GUI-specific Models (SFT)* | | | | | | | | |
| SeeClick (Cheng et al., 2024) | 9.6B | 78.4 | 50.7 | 70.1 | 29.3 | 55.2 | 32.5 | 55.1 |
| UGround (Gou et al., 2025) | 7B | 75.1 | 84.5 | 85.1 | 61.4 | 84.6 | 71.9 | 76.3 |
| OS-Atlas (Wu et al., 2024) | 4B | 87.2 | 59.7 | 72.7 | 46.4 | 85.9 | 63.1 | 71.9 |
| | 7B | 95.2 | 75.8 | 90.7 | 63.6 | 90.6 | 77.3 | 84.1 |
| UI-TARS (Qin et al., 2025) | 2B | 95.2 | 79.1 | 90.7 | 68.6 | 87.2 | 78.3 | 84.7 |
| | 7B | 96.9 | 89.1 | 95.4 | 85.0 | 93.6 | 85.2 | 91.6 |
| | 72B | 94.8 | 86.3 | 91.2 | 87.9 | 91.5 | 87.7 | 90.3 |
| TongUI (Zhang et al., 2025a) | 3B | 94.4 | 79.6 | 92.8 | 75.0 | 87.6 | 77.8 | 85.5 |
| | 7B | 93.1 | 81.5 | 96.4 | 82.9 | 90.2 | 84.7 | 88.7 |
| GUI-Actor (Wu et al., 2025) | 2B | 95.0 | 82.2 | 92.2 | 81.8 | 92.9 | 82.7 | 88.6 |
| | 7B | 96.5 | 84.3 | 91.7 | 84.1 | 93.9 | 82.3 | 89.5 |
| JEDI (Xie et al., 2025) | 3B | 96.6 | 81.5 | 96.9 | 78.6 | 88.5 | 83.7 | 88.6 |
| | 7B | 96.9 | 87.2 | 95.9 | 87.9 | 94.4 | 84.2 | 91.7 |
| *GUI-specific Models (RFT)* | | | | | | | | |
| UI-R1 (Lu et al., 2025) | 3B | 96.2 | 84.3 | 92.3 | 63.6 | 89.2 | 75.4 | 85.4 |
| UI-R1-E (Lu et al., 2025) | 3B | 98.2 | 83.9 | 94.8 | 75.0 | 83.7 | 93.2 | 89.5 |
| SE-GUI (Yuan et al., 2025) | 7B | - | - | - | - | - | - | 90.3 |
| LPO (Tang et al., 2025b) | 8B | 97.9 | 82.9 | 95.9 | 86.4 | 95.6 | 84.2 | 90.5 |
| GUI-G$^2$ (Tang et al., 2025a) | 7B | 98.3 | 91.9 | 95.4 | 89.3 | 94.0 | 87.7 | 93.3 |
| *Ours* | | | | | | | | |
| HyperClick | 3B | 98.6 | 86.3 | 95.4 | 90.6 | 82.2 | 84.7 | 90.6 |
| | 7B | 98.3 | 93.4 | 96.9 | 85.7 | 96.2 | 86.7 | **93.7** |

GUI grounding benchmarks. Detailed experimental results and comparisons with baselines are shown in Table 10.

Table 10: GUI grounding accuracy on the ScreenSpot-Pro (Li et al., 2025) benchmark over the CAD, Development, Creative, Scientific, Office, and OS sub-tasks. **Bold** and underline indicate the best and second-best results.

| Model | Size | CAD | | Development | | Creative | | Scientific | | Office | | OS | | Avg. |
|---|---|---|---|---|---|---|---|---|---|---|---|---|---|---|
| | | Text (197) | Icon (64) | Text (154) | Icon (145) | Text (198) | Icon (143) | Text (144) | Icon (110) | Text (177) | Icon (53) | Text (107) | Icon (89) | |
| *General Models* | | | | | | | | | | | | | | |
| GPT-4o (OpenAI, 2024) | - | 2.0 | 0.0 | 1.3 | 0.0 | 1.0 | 0.0 | 2.1 | 0.0 | 1.1 | 0.0 | 0.0 | 0.0 | 0.8 |
| Claude (Anthropic, 2024) | - | 14.5 | 3.7 | 22.0 | 3.9 | 25.9 | 3.4 | 33.9 | 15.8 | 30.1 | 16.3 | 11.0 | 4.5 | 17.1 |
| Qwen2.5-VL (Bai et al., 2025) | 3B | 9.1 | 7.3 | 22.1 | 1.4 | 26.8 | 2.1 | 38.2 | 7.3 | 33.9 | 15.1 | 10.3 | 1.1 | 16.1 |
| | 7B | 16.8 | 1.6 | 46.8 | 4.1 | 35.9 | 7.7 | 49.3 | 7.3 | 52.5 | 20.8 | 37.4 | 6.7 | 26.8 |
| *GUI-specific Models (SFT)* | | | | | | | | | | | | | | |
| CogAgent (Hong et al., 2024) | 18B | 7.1 | 3.1 | 14.9 | 0.7 | 9.6 | 0.0 | 22.2 | 1.8 | 13.0 | 0.0 | 5.6 | 0.0 | 7.7 |
| SeeClick (Cheng et al., 2024) | 9.6B | 2.5 | 0.0 | 0.6 | 0.0 | 1.0 | 0.0 | 3.5 | 0.0 | 1.1 | 0.0 | 2.8 | 0.0 | 1.1 |
| ShowUI (Lin et al., 2025) | 2B | 2.5 | 0.0 | 16.9 | 1.4 | 9.1 | 0.0 | 13.2 | 7.3 | 15.3 | 7.5 | 10.3 | 2.2 | 7.7 |
| Aria-UI (Yang et al., 2024a) | 25.3B | 7.6 | 1.6 | 16.2 | 0.0 | 23.7 | 2.1 | 27.1 | 6.4 | 20.3 | 1.9 | 4.7 | 0.0 | 11.3 |
| UGround (Gou et al., 2025) | 7B | 14.2 | 1.6 | 26.6 | 2.1 | 27.3 | 2.8 | 31.9 | 2.7 | 31.6 | 11.3 | 17.8 | 0.0 | 16.5 |
| UGround-V1 (Gou et al., 2025) | 7B | 15.8 | 1.2 | 51.9 | 2.8 | 47.5 | 9.7 | 57.6 | 14.5 | 60.5 | 13.2 | 38.3 | 7.9 | 45.2 |
| OS-Atlas (Wu et al., 2024) | 4B | 2.0 | 0.0 | 7.1 | 0.0 | 3.0 | 1.4 | 9.0 | 5.5 | 5.1 | 3.8 | 5.6 | 0.0 | 3.7 |
| | 7B | 12.2 | 4.7 | 33.1 | 1.4 | 28.8 | 2.8 | 37.5 | 7.3 | 33.9 | 5.7 | 27.1 | 4.5 | 18.9 |
| UI-TARS (Qin et al., 2025) | 2B | 17.8 | 4.7 | 47.4 | 4.1 | 42.9 | 6.3 | 56.9 | 17.3 | 50.3 | 17.0 | 21.5 | 5.6 | 27.7 |
| | 7B | 20.8 | 9.4 | 58.4 | 12.4 | 50.0 | 9.1 | 63.9 | 31.8 | 63.3 | 20.8 | 30.8 | 16.9 | 35.7 |
| | 72B | 18.8 | 12.5 | 62.9 | 17.2 | 57.1 | 15.4 | 64.6 | 20.9 | 63.3 | 26.4 | 42.1 | 15.7 | 38.1 |
| TongUI (Zhang et al., 2025a) | 3B | 11.7 | 1.6 | 32.5 | 0.7 | 24.8 | 2.8 | 43.1 | 12.7 | 32.8 | 7.6 | 15.0 | 1.1 | 18.0 |
| | 7B | 17.3 | 9.4 | 40.9 | 3.5 | 31.3 | 7.0 | 50.7 | 12.7 | 45.8 | 13.2 | 28.0 | 6.7 | 24.7 |
| GUI-Actor (Wu et al., 2025) | 2B | - | - | - | - | - | - | - | - | - | - | - | - | 36.7 |
| | 7B | - | - | - | - | - | - | - | - | - | - | - | - | 40.7 |
| JEDI (Xie et al., 2025) | 3B | 27.4 | 9.4 | 61.0 | 13.8 | 53.5 | 8.4 | 54.2 | 18.2 | 64.4 | 32.1 | 38.3 | 9.0 | 36.1 |
| | 7B | 38.0 | 14.1 | 42.9 | 11.0 | 50.0 | 11.9 | 72.9 | 25.5 | 75.1 | 47.2 | 33.6 | 16.9 | 39.5 |
| *GUI-specific Models (RFT)* | | | | | | | | | | | | | | |
| UI-R1 (Lu et al., 2025) | 3B | 11.2 | 6.3 | 22.7 | 4.1 | 27.3 | 3.5 | 42.4 | 11.8 | 32.2 | 11.3 | 13.1 | 4.5 | 17.8 |
| UI-R1-E (Lu et al., 2025) | 3B | 37.1 | 12.5 | 46.1 | 6.9 | 41.9 | 4.2 | 56.9 | 21.8 | 65.0 | 26.4 | 32.7 | 10.1 | 33.5 |
| GUI-R1 (Luo et al., 2025) | 3B | 26.4 | 7.8 | 33.8 | 4.8 | 40.9 | 5.6 | 61.8 | 17.3 | 53.6 | 17.0 | 28.1 | 5.6 | 28.6 |
| | 7B | 23.9 | 6.3 | 49.4 | 4.8 | 38.9 | 8.4 | 55.6 | 11.8 | 58.7 | 26.4 | 42.1 | 16.9 | 31.3 |
| InfiGUI-R1 (Liu et al., 2025b) | 3B | 33.0 | 14.1 | 51.3 | 12.4 | 44.9 | 7.0 | 58.3 | 20.0 | 65.5 | 28.3 | 43.9 | 12.4 | 35.7 |
| GUI-G1 (Zhou et al., 2025) | 3B | 39.6 | 9.4 | 50.7 | 10.3 | 36.6 | 11.9 | 61.8 | 30.0 | 67.2 | 32.1 | 23.5 | 10.6 | 37.1 |
| SE-GUI (Yuan et al., 2025) | 3B | 38.1 | 12.5 | 55.8 | 7.6 | 47.0 | 4.9 | 61.8 | 16.4 | 59.9 | 24.5 | 40.2 | 12.4 | 35.9 |
| | 7B | 51.3 | 42.2 | 68.2 | 19.3 | 57.6 | 9.1 | 75.0 | 28.2 | 78.5 | 43.4 | 49.5 | 25.8 | 47.3 |
| GUI-G$^2$ (Tang et al., 2025a) | 7B | 55.8 | 12.5 | 68.8 | 17.2 | 57.1 | 15.4 | 77.1 | 24.5 | 74.0 | 32.7 | 57.9 | 21.3 | 47.5 |
| *Ours* | | | | | | | | | | | | | | |
| HyperClick | 3B | 43.7 | 23.5 | 62.4 | 20.0 | 50.5 | 12.6 | 55.6 | 30.0 | 63.9 | 37.8 | 41.1 | 20.2 | 41.3 |
| | 7B | 51.3 | 20.3 | 70.2 | 22.1 | 57.6 | 20.3 | 76.4 | 30.9 | 70.1 | 30.2 | 56.1 | 22.5 | **48.2** |

**MMBench-GUI** organizes tasks into a hierarchical structure of basic and advanced instructions. This design enables the systematic evaluation of model performance across varying levels of instruction complexity. Detailed experimental results and comparisons with baselines are shown in Table 11.

Table 11: GUI grounding accuracy on the MMBench-GUI (Wang et al., 2025) benchmark over the Windows, MacOS, Linux, iOS, Android, and Web sub-stasks. **Bold** and underline indicate the best and second-best results.

| Model | Size | Windows | | MacOS | | Linux | | iOS | | Android | | Web | | Avg. |
|---|---|---|---|---|---|---|---|---|---|---|---|---|---|---|
| | | Basic (271) | Adv. (272) | Basic (345) | Adv. (346) | Basic (191) | Adv. (196) | Basic (314) | Adv. (330) | Basic (356) | Adv. (355) | Basic (310) | Adv. (308) | |
| *General Models* | | | | | | | | | | | | | | |
| GPT-4o (OpenAI, 2024) | - | 1.5 | 1.1 | 8.7 | 4.3 | 1.1 | 1.0 | 5.1 | 3.3 | 2.5 | 1.4 | 3.2 | 2.9 | 2.9 |
| Claude (Anthropic, 2024) | - | 1.5 | 0.7 | 12.5 | 7.5 | 1.1 | 0.0 | 13.7 | 10.6 | 1.4 | 1.4 | 3.2 | 2.3 | 4.7 |
| Qwen-Max-VL (Bai et al., 2023) | - | 43.9 | 36.8 | 58.8 | 56.1 | 53.9 | 30.1 | 77.4 | 59.1 | 79.5 | 70.1 | 74.8 | 58.8 | 58.0 |
| Qwen2.5-VL (Bai et al., 2025) | 7B | 31.4 | 16.5 | 31.3 | 22.0 | 21.5 | 12.2 | 66.6 | 55.2 | 35.1 | 35.2 | 40.3 | 32.5 | 33.9 |
| | 72B | 55.7 | 33.8 | 49.9 | 30.1 | 40.3 | 20.9 | 56.1 | 28.2 | 55.6 | 25.4 | 68.4 | 45.8 | 41.8 |
| InternVL3 (Zhu et al., 2025) | 72B | 70.1 | 42.6 | 75.7 | 52.3 | 59.2 | 41.3 | 93.6 | 80.6 | 92.7 | 78.6 | 90.7 | 65.9 | 72.2 |
| *GUI-specific Models (SFT)* | | | | | | | | | | | | | | |
| ShowUI (Lin et al., 2025) | 2B | 9.2 | 4.4 | 24.1 | 10.4 | 25.1 | 11.7 | 29.0 | 19.7 | 17.4 | 8.7 | 22.9 | 12.7 | 16.0 |
| OS-Atlas (Wu et al., 2024) | 7B | 36.9 | 18.8 | 44.4 | 21.7 | 31.4 | 13.3 | 74.8 | 48.8 | 69.6 | 46.8 | 61.3 | 35.4 | 41.4 |
| Aguvis (Xu et al., 2024) | 7B | 37.3 | 21.7 | 48.1 | 33.3 | 33.5 | 25.0 | 67.5 | 65.2 | 61.0 | 51.0 | 61.6 | 45.5 | 45.7 |
| UGround-V1 (Gou et al., 2025) | 7B | 66.8 | 39.0 | 71.3 | 48.6 | 56.5 | 31.1 | 92.7 | 70.9 | 93.5 | 71.0 | 88.7 | 64.6 | 65.7 |
| UI-TARS (Qin et al., 2025) | 72B | 78.6 | 51.8 | 80.3 | 62.7 | 68.6 | 51.5 | 90.8 | 81.2 | 93.0 | 80.0 | 88.1 | 68.5 | 74.3 |
| *Ours* | | | | | | | | | | | | | | |
| HyperClick | 3B | 73.8 | 45.6 | 80.3 | 52.9 | 66.5 | 35.7 | 91.4 | 72.7 | 92.4 | 74.9 | 89.1 | 60.1 | 71.4 |
| | 7B | 82.3 | 61.4 | 82.9 | 67.1 | 66.5 | 48.0 | 94.0 | 82.1 | 95.8 | 85.1 | 93.2 | 85.1 | **79.6** |

**UI-I2E-Bench** introduces implicit instructions that require both semantic understanding and spatial reasoning. Highlights the limitations of direct grounding and encourages models to adopt more sophisticated reasoning. Detailed experimental results and comparisons with baselines are shown in Table 12.

Table 12: GUI grounding accuracy on the UI-I2E-Bench (Liu et al., 2025a) benchmark over the platforms of mobile, desktop, and web with various implicitness. **Bold** and underline indicate the best and second-best results.

| Model | Size | Platform | | | Implicitness | | Avg. |
|---|---|---|---|---|---|---|---|
| | | Mobile (705) | Desktop (519) | Web (253) | Explicit (917) | Implicit (560) | |
| *General Models* | | | | | | | |
| | 3B | 44.5 | 38.7 | 39.9 | 51.4 | 35.8 | 41.7 |
| Qwen2.5-VL (Bai et al., 2025) | 7B | 61.7 | 41.6 | 56.9 | 58.4 | 51.0 | 53.8 |
| | 72B | 55.3 | 47.2 | 49.0 | 49.6 | 52.5 | 51.4 |
| *GUI-specific Models (SFT)* | | | | | | | |
| ShowUI (Lin et al., 2025) | 2B | 53.9 | 30.4 | 29.6 | 51.3 | 35.6 | 41.5 |
| SeeClick (Cheng et al., 2024) | 9.6B | 37.2 | 15.8 | 18.2 | 37.1 | 19.9 | 26.4 |
| Aguvis (Xu et al., 2024) | 7B | 60.3 | 47.6 | 45.1 | 61.1 | 48.4 | 53.2 |
| OmniParser (Wan et al., 2024) | - | 67.6 | 45.5 | 30.8 | 54.3 | 52.4 | 53.1 |
| OmniParser (Yu et al., 2025) | - | 69.4 | 42.4 | 40.7 | 57.0 | 53.5 | 54.8 |
| OS-Atlas (Wu et al., 2024) | 4B | 58.6 | 19.9 | 54.6 | 51.5 | 39.9 | 44.3 |
| | 7B | 68.1 | 48.9 | 52.2 | 63.2 | 55.8 | 58.6 |
| UGround-V1 (Gou et al., 2025) | 2B | 59.9 | 49.5 | 66.4 | 72.9 | 47.9 | 57.4 |
| | 7B | 73.5 | 65.7 | 70.8 | 81.3 | 63.6 | 70.3 |
| | 72B | 78.2 | 74.6 | 74.7 | 84.5 | 71.3 | 76.3 |
| UI-TARS (Qin et al., 2025) | 2B | 66.7 | 54.0 | 62.2 | 74.1 | 54.5 | 62.0 |
| | 7B | 65.7 | 58.0 | 56.5 | 71.4 | 55.3 | 61.4 |
| | 72B | 75.5 | 69.8 | 77.1 | 80.9 | 69.4 | 73.7 |
| UI-I2E-VLM (Liu et al., 2025a) | 4B | 61.4 | 38.9 | 60.9 | 61.9 | 48.3 | 53.4 |
| | 7B | 76.2 | 64.0 | 62.1 | 72.0 | 67.9 | 69.5 |
| *GUI-specific Models (RFT)* | | | | | | | |
| UI-R1 (Lu et al., 2025) | 3B | 67.8 | 46.2 | 58.1 | 67.9 | 52.8 | 58.5 |
| *Ours* | | | | | | | |
| HyperClick | 3B | 77.9 | 59.0 | 81.0 | 81.1 | 66.1 | 71.8 |
| | 7B | 80.4 | 67.5 | 84.2 | 84.8 | 71.4 | **76.5** |

**CAGUI** is a Chinese benchmark for mobile GUI grounding. It emphasizes the grounding of textual elements and functional operations within Chinese-language applications. Detailed experimental results and comparisons with baselines are shown in Table 13.

**UI-Vision** evaluates the generalization of cross-applications in diverse desktop environments. By incorporating previously unseen applications, it tests the model's robustness and adaptability. Detailed experimental results and comparisons with baselines are shown in Table 14.

Table 13: GUI grounding accuracy on the CAGUI (Zhang et al., 2025c) benchmark over the Fun2Point, Text2Point, and Bbox2Text sub-tasks. **Bold** and underline indicate the best and second-best results.

| Model | Size | Fun2Point (1500) | Text2Point (1500) | Avg. |
|---|---|---|---|---|
| *General Models* | | | | |
| GPT-4o (OpenAI, 2024) | - | 22.1 | 19.9 | 21.0 |
| Qwen2.5-VL (Bai et al., 2025) | 7B | 59.8 | 59.3 | 59.6 |
| InternVL2.5 (Chen et al., 2024) | 8B | 17.2 | 24.2 | 20.7 |
| | 26B | 14.8 | 16.6 | 15.7 |
| *GUI-specific Models (SFT)* | | | | |
| OS-Genesis (Sun et al., 2024) | 7B | 8.3 | 5.8 | 7.1 |
| OS-Altas (Wu et al., 2024) | 7B | 53.6 | 60.7 | 57.2 |
| Aguvis (Xu et al., 2024) | 7B | 60.8 | 76.5 | 68.7 |
| UI-TARS (Qin et al., 2025) | 7B | 56.8 | 66.7 | 61.8 |
| *GUI-specific Models (RFT)* | | | | |
| AgentCPM-GUI (Zhang et al., 2025c) | 8B | 79.1 | 76.5 | 77.8 |
| Ours | | | | |
| HyperClick | 3B | 80.9 | 81.2 | 81.0 |
| | 7B | 82.7 | 83.1 | **82.9** |

Table 14: GUI grounding accuracy on the UI-Vision (Nayak et al., 2025) benchmark over the Education (Ed.), Browsers (Br.), Development (De.), Productivity (Pr.), Creativity (Cr.), and Entertainment (En.) subtasks. **Bold** and underline indicate the best and second-best results.

| Model | Size | Setting | | | Category | | | | | | Avg. |
|---|---|---|---|---|---|---|---|---|---|---|---|
| | | Basic (1772) | Functional (1772) | Spatial (1935) | Ed. (642) | Br. (143) | De. (1090) | Pr. (1950) | Cr. (1462) | En. (192) | |
| *General Models* | | | | | | | | | | | |
| GPT-4o (OpenAI, 2024) | - | 1.6 | 1.5 | 1.0 | 1.5 | 0.0 | 2.2 | 1.1 | 0.8 | 4.2 | 1.4 |
| Gemini-1.5-pro (Team et al., 2024) | - | 0.8 | 0.3 | 0.6 | 0.5 | 0.6 | 0.9 | 0.5 | 0.4 | 0.0 | 0.6 |
| Claude (Anthropic, 2024) | - | 9.5 | 7.7 | 7.6 | 6.1 | 9.8 | 8.0 | 9.4 | 7.7 | 8.3 | 8.3 |
| Qwen2.5-VL (Wang et al., 2024a) | 7B | 1.2 | 0.8 | 0.5 | 0.5 | 0.0 | 1.2 | 0.9 | 0.5 | 1.0 | 0.9 |
| InternVL2.5 (Chen et al., 2024) | 8B | 2.5 | 2.8 | 1.0 | 1.1 | 7.0 | 3.0 | 1.8 | 1.2 | 5.2 | 2.1 |
| MiniCPM-V (Yao et al., 2024) | 8B | 7.1 | 5.3 | 1.5 | 3.0 | 16.8 | 5.4 | 3.8 | 2.1 | 13.0 | 4.3 |
| *GUI-specific Models (SFT)* | | | | | | | | | | | |
| CogAgent (Hong et al., 2024) | 9B | 12.0 | 12.2 | 2.6 | 8.7 | 11.2 | 8.6 | 10.3 | 5.6 | 15.6 | 8.9 |
| SeeClick (Cheng et al., 2024) | 9.6B | 9.4 | 4.7 | 2.1 | 4.2 | 13.3 | 7.3 | 4.3 | 4.0 | 11.0 | 5.4 |
| AriaUI (Yang et al., 2024a) | 25.3B | 12.2 | 14.0 | 4.0 | 9.0 | 18.9 | 11.2 | 10.4 | 6.5 | 19.3 | 10.1 |
| ShowUI (Lin et al., 2025) | 2B | 8.1 | 7.7 | 2.1 | 3.7 | 13.3 | 7.5 | 6.5 | 2.5 | 15.6 | 5.9 |
| OS-Atlas (Wu et al., 2024) | 7B | 12.2 | 11.2 | 3.7 | 8.7 | 16.8 | 10.3 | 9.2 | 5.6 | 16.2 | 9.0 |
| UGround-V1 (Nayak et al., 2025) | 7B | 15.4 | 17.1 | 6.3 | 10.4 | 28.7 | 17.5 | 12.2 | 8.6 | 18.2 | 12.9 |
| | 72B | 27.9 | 26.7 | 14.9 | 22.4 | 35.7 | 27.6 | 21.6 | 18.3 | 38.0 | 23.2 |
| Aguvis (Xu et al., 2024) | 7B | 17.8 | 18.3 | 5.1 | 13.1 | 30.8 | 17.1 | 12.1 | 9.6 | 24.0 | 13.7 |
| UI-TARS (Qin et al., 2025) | 7B | 20.1 | 24.3 | 8.4 | 14.2 | 35.0 | 19.7 | 18.3 | 11.1 | 38.5 | 17.6 |
| | 72B | 31.4 | 30.5 | 14.7 | 24.8 | 40.5 | 27.9 | 26.8 | 26.8 | 17.8 | 25.5 |
| *Ours* | | | | | | | | | | | |
| HyperClick | 3B | 28.7 | 24.4 | 6.8 | 19.6 | 30.8 | 20.6 | 21.1 | 12.7 | 40.6 | 19.6 |
| | 7B | 35.3 | 32.1 | 11.0 | 24.3 | 47.6 | 26.5 | 27.1 | 18.3 | 50.0 | **25.7** |

