# OpenReview forum: "HyperClick: Advancing Reliable GUI Grounding via Uncertainty Calibration"
_ICLR.cc/2026/Conference — ICLR 2026 Conference Withdrawn Submission_

### Official Review · Reviewer_FP79 · 2025-10-21

**Soundness:** 2
**Presentation:** 2
**Contribution:** 2
**Rating:** 4
**Confidence:** 4

**Summary:**

This paper aims at better GUI grounding more reliably. It proposes a dual reward mechanism, combining a binary reward with a Gaussian–based reward to reduce overconfidence. Its novelty against other similar works lies in its awareness at confidence. But I really doubt the novelty and the contribution of **Gaussian-based reward** and the **confidence** to performance gains. So I think current version is below the ICLR conference's standard and I tend to rejection.

**Strengths:**

1. The main result benchmarks are sufficient, even including CAGUI and MMBenchGUI which are not so widely used.
2. The performance achieves SOTA among several open-source models reported.
3. I personally like the starting point of this paper: using Probabilistic Confidence and Verbalized Confidence as preliminary studies.

**Weaknesses:**

1. About **Novelty**. The biggest weakness lies in novelty.  This paper incorporates Gaussian–based term in GRPO rewards to reduce overconfidence. However, the **confidence** and **Gaussian–based reward** are not so timely now. For **confidence**, Visual-RFT [1] aims at reducing overconfidence by introducing $R_{conf}$. The ideas are similar with HyperClick: for successfully matched boxes, the higher the confidence, the better. For **Gaussian-based reward**, GUI-G$^2$ [2] first proposes Gaussian-based Point Rewards and Gaussian-based Coverage Rewards in GUI grounding. In my point, HyperClick uses a binary accuracy reward like UI-R1 [3] and a Gaussian-based confidence reward followed followed by GUI-G$^2$ [2]. The reward designs are out of time to some extend.
2. About **Confidence reward**. Besides novelty, the paper lacks ablations about confidence reward. Current experiments doesn't support the necessity of **Gaussian-based** in confidence reward. Second, you listed two kinds of confidence at beginning, but there lacks further analysis or experiments. The storytelling feels somewhat inconsistent and fragmented. For example, what's the results of using Probabilistic Confidence, Verbalized Confidence, or combining them two into the confidence reward.



[1] Visual-RFT: Visual Reinforcement Fine-Tuning

[2] GUI-G2: Gaussian Reward Modeling for GUI Grounding

[3] UI-R1: Enhancing Efficient Action Prediction of GUI Agents by Reinforcement Learning

**Questions:**

1. More discussions about novelty.
2. More ablations about confidence reward design.
3. More demonstrations about the role and necessity of confidence reward. For example, (1) more results of incorporating Gaussian-based reward into $R_{correctness}$ instead of $R_{confidence}$; (2) result comparison with GUI-G2's reward design under the same training setting. In my point, improving accuracy is more prior to reducing overconfidence.
5. More results tested on all benchmarks would be better instead of '-'.

---

### Official Review · Reviewer_YdFo · 2025-10-21

**Soundness:** 4
**Presentation:** 3
**Contribution:** 3
**Rating:** 6
**Confidence:** 3

**Summary:**

This paper proposes a plug-and-play training paradigm named HyperClick for GUI grounding, aiming at enhancing reliable GUI grounding through uncertainty calibration. HyperClick introduces correctness reward and confidence reward, jointly optimizing grounding accuracy and confidence reliability while fostering introspective self-criticism. Experimental results on seven challenge benchmarks show that HyperClick outperforms baselines.

**Strengths:**

1. GUI grounding is the fundamental GUI adaption for GUI agents that enables them to identify GUI elements for specific user command. However, current models lack self-awareness of their capability boundaries, leading to overconfidence and unreliable predictions. Enhancing the reliability for GUI grounding is critical for the robustness of GUI agents.
2. The proposed HyperClick introduces correctness reward and confidence reward, which are designed to jointly optimize grounding accuracy and confidence reliability while fostering introspective self-criticism. This is a novel approach to improve the reliability of GUI grounding.
3. Experimental results on seven grounding benchmarks demonstrate that HyperClick outperforms baselines, indicating its effectiveness in improving the reliability of GUI grounding.

**Weaknesses:**

1. The utilization of a Gaussian representation was previously introduced in GUI-G2, which makes parts of this methodology appear similar. A more explicit differentiation from GUI-G2 is needed to better clarify the novelty of HyperClick.
2. The ablation in Table 3 omits the vanilla baseline where only $R_{format}$ is adopted, which may provide a better understanding of the impact of the proposed methods.
3. The 3B models surprisingly outperforms 7B models in Table 2, which is counterintuitive. Further analysis is needed to explain this result and ensure fair comparison.
4. Typo errors:
   - In Section 4.3 Comparisons with Baselines, Table 4 should be Table 1.

**Questions:**

1. Highlight the novelty of this paper compared to GUI-G2.
2. Add vanilla baseline where only $R_{format}$ is adopted in Table 3.
3. Explain why the 3B models outperform 7B models in Table 2.

---

### Official Review · Reviewer_vwD3 · 2025-10-28

**Soundness:** 1
**Presentation:** 2
**Contribution:** 2
**Rating:** 2
**Confidence:** 2

**Summary:**

The authors focus on GUI grounding, highlighting and empirically showing that prior approaches are frequently overconfident in their predictions. They aim to reduce this overconfidence with HyperClick, a method that utilizes uncertainty calibration to improve performance and better calibrate confidence. They evaluate their approach on a suite of seven GUI grounding benchmarks, showing high accuracy on all tasks.

**Strengths:**

- The motivation for this work is strong, modeling uncertainty in GUI grounding is an important problem, especially as GUI agents are given more access.

**Weaknesses:**

- The motivation for additional datasets (MMG, I2E, CAG, UIV) is not clear, especially given that the authors have only reported results for 2/36 RFT model-dataset pairs (which seem to outperform SFT models on average).
- It is unclear whether the proposed approach outperforms GUI-G2. GUI-G2 is only evaluated on three of the seven datasets and their performance is within a percent on these three datasets.
- Additionally, one of the main benefits of the confidence loss seems to be the improved calibration of confidence scores. Why were the subset of models in Table 2 selected from the full set in Table 1? Why were RFT models not evaluated? Why does the 3B model have higher average precision at lower thresholds?

**Questions:**

- I was hoping the authors could help interpret the dataset metrics. What does a percentage point correspond to in these datasets? From my understand ScreenSpot-Pro has 1581 natural language instructions, does this mean that the best result from Table 3 follows 11 more instructions correctly than the baseline approach with only the correctness reward? Is this a meaningful improvement?
- Do the authors have intuition for why the 1/6 result is lower than 0 in Table 4?
- "Introducing the confidence reward alone already achieves stronger performance (48.0%), while the combination of correctness and confidence rewards further increases the precision to 48.2%." Could the authors clarify this point, it does not seem aligned with the Table, as there is no confidence reward alone.
- Could additional results be collected showing the impact of the confidence reward on the calibration of the confidence scores? For example, can HyperClick be run without the confidence reward on the evaluation from Table 2?
- Typo line 397: Rcorectness

---

### Official Review · Reviewer_Zwy1 · 2025-10-31

**Soundness:** 3
**Presentation:** 3
**Contribution:** 2
**Rating:** 4
**Confidence:** 3

**Summary:**

The paper introduces HyperClick, a framework to address the poor confidence calibration of GUI grounding models. The authors identify that existing models are often overconfident, producing unreliable predictions that are detrimental in GUI tasks.

**Strengths:**

- Well-Posed and Important Problem. GUI grounding is a key task, and overconfidence is a key obstacle.

- Extensive and sound experiments. Many ablations on the algorithm and models are conducted to verify the design of HyperClick.

**Weaknesses:**

- Unclear contribution of modules. Though Table 3 and 4 list some ablations of the algorithm, they showed very small differences, especially in the reward configurations. The key contribution of this work, the confidence reward, only brings a marginal 0.5% improvement.

- The confidence assumption: HyperClick assumes the model to be the most confident when predicting the center of the bounding box, and decays in a Gaussian function. Some data labels (especially human-annotated ones) may not be at the exact center, which undermines the claim. An experiment can make it more convincing.


### Typos and minor presentation problems

1. A redundant comma after "Table 4" in Section 4.4
2. In the caption of Figure 3, "darker" color can be ambiguous.

**Questions:**

See Weaknesses.

---

### Official Review · Reviewer_t3By · 2025-11-01

**Soundness:** 3
**Presentation:** 3
**Contribution:** 3
**Rating:** 2
**Confidence:** 4

**Summary:**

A new SOTA system is presented for the task of "GUI grounding."
The system input is (screenshot, natural language instruction) where the instruction identifies a target element such as a GUI button.
The system must predict a click location (x,y) within the target's bounding box.

The authors augment this primary task with an auxiliary task: they ask the system to also predict a centrality c that assesses how close the predicted (x,y) is to the center of the target.

Primary reward: The predicted (x,y) gets reward 1 or 0 according to whether (x,y) falls within the target's bounding box.
Auxiliary reward: The predicted c gets reward ≤ 0 for getting the centrality wrong.  This penalty is the squared L2 distance from the true centrality of (x,y).

The specific predictive model is an LLM (Qwen 3B or 7B) that verbalizes (x,y,c) as decimal numerals.  The authors fine-tune it via GRPO to output (x,y,c) that achieve high total reward (primary + auxiliary).

The hope is that there will be positive transfer from the auxiliary task: The system will learn to *internally* predict the true centrality because this is necessary to improve the auxiliary reward, and that could help it achieve a high primary reward by learning to avoid predictions that it thinks have poor centrality.

The true centrality is defined as exp(-distance from the center), where distance is Euclidean but measured in a coordinate system where the bounding box has height and width of α.  This is 1 if (x,y) is exactly at the center.  (Exception: the centrality is forced to 0 if (x,y) is outside the bounding box.)

**Strengths:**

Significance: At line 370, Table 1 shows SOTA performance of the 7B model on 6 of 7 benchmarks, and second-best performance on the remaining benchmark.  Unfortunately, it is not clear to me that this is because of the paper's contribution.  It might just be the strength of the underlying model (see questions in later section).

Table 2 and Figure 3 suggest that the model's predicted confidences are informative and could be useful downstream.

Originality: It's a nice idea to use some kind of uncertainty quantification as an auxiliary task that helps representation learning.  But I wouldn't say that is original to this paper.  It's very common to build a system that predicts a probability distribution over outputs rather than just predicting the output directly.  The most basic example is heteroscedastic regression; for an AI reference, consider [Kendall & Gal (2017)](https://arxiv.org/abs/1703.04977).  The submitted paper uses rewards (basically just one-step rewards), so perhaps distributional RL work is even more relevant, e.g., [Dabney et al. (2017)](https://arxiv.org/abs/1710.10044).

**Weaknesses:**

The method feels ad hoc.  Why not start with ordinary uncertainty quantification methods?

The paper's terminology and writing are often imprecise.  I don't think c is really a "confidence" in the usual sense.  I don't think a low L2 error in predicting c is really "calibration" in the usual sense.  (Calibration would involve mapping the predicted c values to actual error rates on the primary task, e.g., using isotonic regression or Platt scaling.)  Nor should that L2 error be called a Brier score (line 250) since Brier scores are for probabilistic classifiers, not regressors.

**Questions:**

Line 399 suggests that the auxiliary reward - which is advertised as the paper's main contribution - only increases the accuracy from 47.7% to 48.2% on the SSP benchmark.  Is that improvement statistically significant?  Do you also see improvements on the 6 other datasets?  In other words, does the strong performance in Table 1 really arise from training with auxiliary reward, or does it just arise from using the Qwen2.5-VL-Instruct models?

The results in Table 2 are hard to interpret because it's not clear how many examples achieve, say, conf=90.  The confidence levels here are weird numbers that are internal to the proposed HyperClick system.  So it would be better to report precision at different coverage levels: how accurate is the system on its most confident 100%, 90%, 80%, ... of the examples?

Did I characterize your method correctly at the start of this review?  And can you explain line 317?  "This enables the policy to penalize overconfident errors while reinforcing well-calibrated clicks."  (That sentence doesn't match my understanding of your method.)

What would happen if you did something simpler in place of your confidence reward (6) (which draws on preceding equations)?

* In the confidence estimation literature, I think "confidence" usually refers to estimated probability that the output is exactly correct.  I expected you would ask *the model* to predict a probability distribution over the true bounding box.  (So I was confused when that wasn't what your Gaussians were doing ...)  This would then allow you to obtain a "confidence" in the traditional sense above, namely the probability that the system's output (x,y) falls in the unknown bounding box.  Alternatively, you could just predict the confidence directly as a probability.  Would these methods work?

* I don't think I've seen the term "confidence" used for continuous predictions like (x,y).  However, you could treat 1 minus the traditional confidence as the expected 0-1 loss.  The natural generalization to continuous predictions would be to instead measure, say, the expected L2 loss of (x,y).  Would that work?

* Your idea of predicting the centrality of (x,y) is also a reasonable heuristic.  But would a simpler definition of centrality have worked, such as the margin?  (That is, the distance of (x,y) from the edge of the bounding box, signed positive or negative according to whether (x,y) is inside or outside.  Again, this would be measured in a coordinate system where the bounding box has height and width of α.)

* One consequence of your actual centrality definition (3) is that two predictions that are both far from the center will both have similarity ≈ 0, even if one is much farther than the other.  In other words, the exponentiation results in treating all large distances as nearly equal to one another.  Why is that desirable?  (It is not true for L2 loss or margin, as suggested in the two previous paragraphs.)

---

### Note · Authors · 2025-11-19

I have read and agree with the venue's withdrawal policy on behalf of myself and my co-authors.